# Multimodal Resonances of a Rectangular Planar Dielectric Elastomer Actuator and Its Application in a Robot with Soft Bristles

**DOI:** 10.3390/biomimetics9080488

**Published:** 2024-08-13

**Authors:** Yangyang Du, Xiaojun Wu, Dan Wang, Futeng Zhao, Hua Hu

**Affiliations:** 1School of Mechanical and Electrical Engineering, Xi’an University of Architecture and Technology, Xi’an 710049, China; dyy236scu@163.com (Y.D.); danwang@xauat.edu.cn (D.W.); 2Department of Mechanical and Electrical Engineering, Yuncheng University, Yuncheng 044000, China; huhua_2011@163.com; 3Research Centre for Medical Robotics and Minimally Invasive Surgical Devices, Shenzhen Institutes of Advanced Technology (SIAT), Chinese Academy of Sciences, Shenzhen 518055, China; ft.zhao@siat.ac.cn

**Keywords:** dielectric elastomer actuator, multimodal resonance, out of plane, rotation vibration

## Abstract

Inspired by the fact that flying insects improve their power conversion efficiency through resonance, many soft robots driven by dielectric elastomer actuators (DEAs) have achieved optimal performance via first-order modal resonance. Besides first-order resonance, DEAs contribute to multiple innovative functions such as pumps that can make sounds when using multimodal resonances. This study presents the multimodal resonance of a rectangular planar DEA (RPDEA) with a central mass bias. Using a combination of experiments and finite element modeling (FEM), it was discerned that under a prestretch of 1.0 × 1.1, the first-, second-, and third-order resonances corresponded to vertical vibration, rotation along the long axis, and rotation along the short axis, respectively. In first-order resonance, superharmonic, harmonic, and subharmonic responses were activated, while only harmonic and subharmonic responses were observed in the second- and third-order resonances. Further investigations revealed that prestretching tended to inhibit third-order resonance but could elevate the resonance frequencies of the first and second orders. Conveniently, both the experimental and FEM results showed that the frequencies and amplitudes of the multimodal resonances could be tuned by adjusting the amplitudes of the excitation signals, referring to the direct current (DC) amplitude and alternating current (AC) amplitude, respectively. Moreover, instead of linear vibration, we found another novel approach that used rotation vibration to drive a robot with soft bristles via hopping locomotion, showcasing a higher speed compared to the first-order resonance in our robot.

## 1. Introduction

Soft robots, powered by artificial muscles, such as dielectric elastomer actuators (DEAs), shape memory polymers, and memory alloys, have garnered increasing attention in recent decades due to their inherent compliance [1,2,3]. Among these robots, DEAs stand out due to their fast responses, large strain areas, and high electromechanical efficiency [4,5,6]. As a result, robots driven by DEAs have demonstrated a diverse range of functions, such as jumping [7], hopping [7,8,9], crawling [10,11,12], flying [13], rolling [14], pumping [15,16,17], and grabbing [18]. These functions are contingent on the deformation mechanisms of DEAs.

When a DEA undergoes charge-attraction-induced Maxwell stress, the membrane expands to cover a larger area and becomes thinner. Under a constant or slowly varying voltage, DEAs deform in either a static or quasi-static manner. However, when subjected to a periodic excitation signal, the deformation transitions to a dynamic response. Since DEAs can be conceptualized as systems comprising springs and damping, a dynamic response can manifest as resonance, enabling a more substantial power output. Furthermore, inspired by species of flying insects that utilize resonance to amplify their strokes and reduce their power consumption, many bioinspired robots driven by DEAs, such as flying [13,19], crawling [9,20], and pumping robots [15,21], have been granted maximized or optimized performances at their first-order resonance.

However, besides first-order resonance, DEAs can also be interpreted as shell structures actuated by Maxwell stress, suggesting that DEAs can exhibit multimodal resonance and more complex response behaviors. Several studies have observed the multimodal resonances of DEAs and their applications. For instance, Jian Zhu et al. used numerical simulations to predict the presence of multimodal resonance in a hemispheric DEA under pressure [22]. Similarly, Chao Tang et al. uncovered the multimodal resonance of a circular planar DEA without a central mass bias, merging insights from numerical simulations, finite element modeling (FEM), and experiments [20]. They utilized the first-order modal resonance, which displayed a linear vibration perpendicular to the membrane surface, to drive an anisotropic frictional crawling robot [23]. Chongjing Cao et al. experimentally found that a circular planar DEA with a central mass bias exhibited multimodal resonance, with the second-order modal resonance being rotational vibration [24]. Furthermore, Sebastian Gratz-Kelly et al. designed a multifunctional conical DEA integrated with a negative stiffness structure. Attributed to its swift response frequency, this actuator can achieve a linear-actuation pump motion at the first-order resonance and can generate an audible sound at the higher-order resonance [25]. These findings underscore the significance and intrigue of studying the multimodal resonances of DEAs.

Besides the multimodal responses of the above circular, conical, and spherical DEAs, there is a noticeable gap concerning the multimodal resonances of DEAs with alternative shapes, such as rectangular ones. Focusing on rectangular planar DEAs (RPDEAs), several studies have investigated their in-plane deformation, including the static, quasi-static, and low-frequency dynamic responses, by either theory or experiments [26,27,28,29,30]. Similar to other hyperelastic polymer films with inherent nonlinearity [31,32], the RPDEA should also exhibit multimodal resonances, transcending mere in-plane deformation. Nonetheless, based on our current understanding, few works have targeted multimodal resonances, which may be attributed to the complexities presented by its multi-axis structures and inhomogeneous deformation.

Consequently, in this work, we endeavored to uncover the multimodal responses of an RPDEA with a mass bias using a combination of experiments and FEM, including the first three orders of modal resonance, which corresponded to vertical vibration, rotation along the long axis, and rotation along the short axis, respectively. Furthermore, we found that the rotation vibration could lead to a robot with soft bristles crawling directionally via hopping locomotion, which exhibited more advantages in moving speed than the vertical vibration. In addition, rather than the vertical vibration, rotation-vibration-induced hopping locomotion is a new approach to driving a robot with soft bristles, expanding the actuation modes of soft robotics.

The rest of this paper is organized as follows: Section 2 first introduces the design and fabrication procedures of the RPDEA; secondly, it outlines the finite element modeling (FEM) approach adopted for multimodal analysis; finally, it describes the dynamic response testing method. Section 3 (i) uncovers the multimodal dynamic responses of the RPDEA through both experiments and FEM; (ii) studies the effects of the prestretch ratio and excited electric field intensity on the multimodal resonances; and (iii) demonstrates and compares the crawling performances driven by these three resonances. Section 4 concludes this work and proposes future work.

## 2. The Structure, FEM, and Test of the RPDEA

### 2.1. The Structure and Fabrication

The RPDEA comprises a rigid frame, a DE membrane sandwiched between distributed rectangular compliant electrodes on both sides, and a central mass bias (vibrator), as depicted in Figure 1a,b. The Maxwell stress, which caused by the attraction of positive and negative charges located at the two sides of dielectric elastomer, respectively, is periodic under a sinusoidal voltage. The periodic Maxwell stress results in the thickness of the elastomer shrinks in the cycle and the corresponding out-of-plane deformation of RPDEA. The vibrator weighs approximately 0.49 g, and its contour is 10 × 10 × 4.1 mm^3^. The rigid frame has an outer contour of 76 × 38 mm^2^ and an inner dimension of 62 × 24 mm^2^. It is crafted from an acrylic plate with thickness of 2 mm using a laser cutter. We choose the silicone elastomer, Elastosil 2030 (with a thickness of 100 μm from Wacker Chemie AG, Munich, Germany) as the DE membrane for its lower viscoelasticity [33]. All components, namely the DE, vibrator, and frame, are bonded together using double-sided tapes.

The compliant electrode [24], formulated by mixing carbon particles with vegetable oil in a 1:8 weight ratio, is delicately hand-brushed onto the DE membrane with each area of 25 × 20 mm^2^. To prevent tension loss, the DE is stretched lengthwise, whereas its width retains the original stretch ratio of 1. This process is achieved by the setup shown in Figure 1c, mirroring the device used for VHB membrane biaxial stretching.

### 2.2. The Process of FEM

We employed the FEM to explore the multimodal resonances (from the first to the third order) of the RPDEA. The mechanical attributes of the silicone membrane, Elastosil 2030, were ascertained using a uniaxial tensile test (Mark 10, ESM303). These results were subsequently integrated into the FEM using the Yeoh hyperelasticity model for data fitting. The Yeoh hyperelasticity model is described as
(1)$$\begin{array}{c} W\left(\lambda_{1},\lambda_{2}\right)=\frac{C_{1}}{2}\left(\lambda_{1}^{2}+\lambda_{2}^{2}+\lambda_{1}^{-2}\lambda_{2}^{-2}\right)+\frac{C_{2}}{2}{\left(\lambda_{1}^{2}+\lambda_{2}^{2}+\lambda_{1}^{-2}\lambda_{2}^{-2}\right)}^{2}\\ +\frac{C_{3}}{2}{\left(\lambda_{1}^{2}+\lambda_{2}^{2}+\lambda_{1}^{-2}\lambda_{2}^{-2}\right)}^{3}+\overset{3}{\underset{i=1}{\sum }}\frac{1}{D_{i}}{\left(J_{el}-1\right)}^{2i} \end{array}$$
where $\lambda_{1}$ and $\lambda_{2}$ are the stretch in the width and the length of the membrane, respectively. $C_{1}$, $C_{2}$, and $C_{3}$ describe the parameters of shear characteristics, $J_{el}$ is the elastic volume ratio, and $D_{i}$ indicates the parameters of material compressibility. According to the fitting result, Table 1 shows the above parameters of silicone membrane, which are close to those in [34].

To avoid the loss of tension, the DE membrane is prestretched before being glued with the rigid frame and the mass bias in the experiment. Thus, in the simulation, the membrane is also prestretched through the load command in the FEM software. Due to the hyperelasticity, the membrane is meshed with the element C3D8R (an 8-node linear brick, reduced integration, hourglass control), whose initial size is 0.2 × 0.2 × 0.1 mm^3^. After prestretching, the membrane is tied with the mass bias in another model which inherits the prestretched state. The ‘Linear perturbation—Frequency’ and ‘Modal dynamics’ steps are used to identify the multimodal resonance frequency and amplitude, respectively.

### 2.3. The Test Method

We used a bias AC signal to test the dynamic response of the RPDEA while conducting both side compliant electrodes synchronously. The test setup is illustrated in Figure 2, where RPDEA is positioned on an optical table, and a laser displacement sensor (LK-G152 and LKGD500 from Keyence, Osaka, Japan) is used to gauge the displacement of the vibrator. To swiftly pinpoint the resonant frequency, we utilized the analog sweep frequency signal which was then input to high voltage amplifiers (HVA, 10/40A-HS, TREK, New York, USA) via Data Acquisition (DAQ). The HVA subsequently applied the AC voltage (*V_in_*) to the DE membrane. To avoid AC voltage lower than zero [24], the *V_in_* is set as
(2)$$V_{in}=0.5E\left(1+sin2\pi \Omega t\right)T_{0}/(\lambda_{1p}\lambda_{2p})$$
where *E* is the nominal peak electric field intensity; *Ω* is the frequency of signal input to HVA; *T*_0_, *λ*_1*p*_, and *λ*_2*p*_ are the initial thickness and the prestretched ratio of the DE in width and length, respectively. It should be pointed out that this equation is based on an assumption that the DE is approximatively incompressible [24].

The frequency sweep results can identify the resonance region but fail in determining the stable response. In applications, the stable maximum resonant amplitude is selected to maximize robot performance, such as those of flying [13,19] and crawling [9,20] robots. Therefore, in this work, we focused on the maximum amplitude of each modal resonance.

Due to the complex dynamic behavior of RPDEA, we employed two methods to ascertain the stable resonance under a fixed excited frequency. Method I: for the linear actuation, the light-spot of laser displacement sensor is centered on the mass bias. Method II: for the rotational vibration, the light-spot is placed on the edge of vibrator to measure response frequency and estimate the amplitude. The rotational amplitude is then measured from snapshots captured by a high-speed camera at an optimal viewing angle.

## 3. Results

### 3.1. Identification of the Multimodal Resonance

Figure 3 presents the simulation outcomes for the RPDEA subjected to a prestretch of 1.0 × 1.1: Figure 3a depicts the state after prestretching; Figure 3b–d illustrate the resonant behavior corresponding to the first-order, second-order, and third-order modals. With the reference length of silicone membrane at 56.4 mm, the notable displacement of 2.82 mm in the U1 direction (Figure 3a) confirms that *λ*_2_ = 1.1. And the rectangular fringes indicate the homogeneous deformation across the width and thickness.

According to Figure 3b–d, the first-order resonance of the RPDEA manifests as a linear actuation, perpendicular to the membrane surface. In contrast, both the second-order and third-order resonances display rotational vibrations. Specifically, the second-order rotates along the long axis, while the rotation of the third-order is perpendicular to the long axis yet parallel to the short axis. Note that the frequency goes up as the resonance order increases. However, owing to the frames per second (FPS) limitations (≤1300 fps) of the high-speed camera, our analysis excludes the resonances beyond the third order.

To validate these findings, the frequency sweep test under the excited signal, following Equation (2), is applied on the RPDEA. Different to the stable solution identified by the FEM, due to the nonlinear force–displacement relationship of the RPDEA and lacking any coupled elastic units, potential stable solutions are manifold under the frequency sweep. To further elucidate the resonances with stable solution, we employ both a forward sweep (0 to 250 Hz at 1 Hz/s) and a backward sweep (250 to 0 Hz at 1 Hz/s). Figure 4 outlines the results for the RPDEA with prestretch of 1 × 1.1. In the figure, seven prominent peaks demarcated by dotted lines represent the stable solution position of these resonances. It is worth noting that peaks without overlap between the forward sweep and the backward sweep are disregarded due to their instability; that is, these response areas cannot be activated by fixed-frequency excited signals [24]. To discern the resonance modals, we utilize a high-speed camera to monitor the vibration behavior around the peaks, as displayed in Figure 5. Within the 0 Hz to 250 Hz range, three distinct modals manifest: modal 1 exhibits vertical vibration; modal 2 and modal 3 display rotational vibrations. Similar to the FEM results, modal 2 rotates parallel to the long axis, while modal 3 rotates in alignment with the short axis. Note that due to the errors in manual assembly and a slight asymmetry in the compliant electrodes, Figure 5d shows an imperfection where the rotation axis does not perfectly align with the length direction.

The above results confirm the existence of multimodal resonances in RPDEA. Notably, there is a marked discrepancy between the excited frequency and the simulation response frequency across these resonances. For example, in modal 1, the resonant frequency of FEM is 51.2 Hz, while the frequency of the excited signal in the experiment has three values: 96 Hz, 47 Hz, and 23 Hz. To elucidate this phenomenon, the subsequent analysis delves into subharmonic, harmonic, and superharmonic responses.

Different to the modal 2 and modal 3 resonances, modal 1 features three pronounced peaks, as shown in Figure 4. To further distinguish these three peaks, we employed a discrete Fourier transform (DFT) to determine the relationship between response frequency and the excited signal frequency, as shown in Figure 6. Because of the inherent nonlinearity of RPDEA, though the excited signal frequency varies, the dynamic response ω of the modal 1 resonance consistently hovers around approximately 47.5 Hz. Therefore, the first, second, and third peaks of modal 1 resonance marked in Figure 4 correspond to the superharmonic, harmonic, and subharmonic responses, respectively. Because of the nonlinear material damping and structural nonlinearity, the resonant amplitude goes up nonlinearly, although with the excited frequency ascends twofold.

Although modal 2 and modal 3 resonances are evident when merging insights from Figure 4 and Figure 5, pinpointing the relationship between their response frequency and the excited signal frequency remains challenging. To address this, we utilize a laser displacement sensor to gauge the displacement of the vibrator margin at a fixed signal frequency, as indicated by points 1 and 2 in Figure 5a. Figure 7a,b demonstrate the harmonic resonance of modal 2 and modal 3, where the response frequency is same as the excited signal frequency. Meanwhile, as the response frequency is approximatively half of the excited frequency, Figure 7c and d confirm the subharmonic resonances of modal 2 and modal 3. However, different to modal 1, the superharmonic resonances of the latter two are scarcely observed in the experiments. This anomaly reveals that the high-order resonances are more difficult to be activated than the low-order resonances.

Similar to the circular DEA with a central mass bias [24], the RPDEA also manifests significant nonlinearity in its modal 1 resonance under different frequency sweep direction. In the forward-sweep test, significant amplitudes only emerge at the resonance point, plummeting sharply afterward. And especially in the subharmonic response, the peak amplitude can soar to 5.6 mm. However, during the backward sweep, the amplitude jumps from a minimal value and then reduces as the frequency continues to decrease. But the peak amplitude is notably lower in the backward sweep than in the forward, only reaching about 1.78 mm. Due to the fact that the input power escalates as the excited signal frequency heightens, both sweep results show that the first peak (superharmonic) has the lowest amplitude, whereas the third peak (subharmonic) is the highest.

Likewise, in the modal 2 and modal 3 resonances, there also exists an amplitude difference between Figure 7a,c and between Figure 7b,d. The former two, although excited by a higher *E*, have a lower amplitude than that of the latter two. These observations underscore that the subharmonic responses also surpass the harmonic responses in the rotational vibration amplitude.

The above experiments verify the ability of Maxwell stress to activate the multimodal resonance of RPDEA. Based on the observed vibrational motion, in the rest of this paper, the following are true: modal 1 resonance will denote vertical vibration; modal 2 resonance implies rotation along the long axis; modal 3 resonance represents rotation along the short axis.

### 3.2. Effects of the Parameters

In this section, the effects of prestretch variation *λ*_2_ and the excited signal (the nominal peak electric field intensity *E*) on the multimodal of RPDEA are investigated. Firstly, *λ*_2_ was studied, set as 1.1, 1.2, or 1.3, respectively.

#### 3.2.1. The Prestretch Ratio *λ*_2_

As Figure 4 has presented, the frequency sweep result for *λ*_2_ = 1.1; Figure 8 only showcases the outcomes for *λ*_2_ = 1.2 and 1.3 under *E* = 45 MV/m.

According to Figure 8a,b, the superharmonic, harmonic, and subharmonic responses of modal 1 resonance are activated under *λ*_2_ with 1.2 and 1.3. However, for modal 2 resonance, different to *λ*_2_ = 1.1, only the subharmonic response is apparent in these two RPDEAs. And when *λ*_2_ goes up to 1.3, in Figure 8b, we can hardly observe the modal 3 resonance. It should be pointed out that each of these responses has been verified through careful observations using the high-speed camera.

The absence of modal 3 in *λ*_2_ = 1.3 RPDEA can be attributed to a decrease in the effective mass. This inference is supported by the data presented in Figure 9, which are extracted from the ‘Job-Monitor’ in the FEM software. Further, based on the trends illustrated in Figure 9, it appears that there exists a threshold for the effective mass of the rotational vibration, situated between 6 g and 8 g. This phenomenon suggests that increasing the prestretch may inhibit the excitation of higher-order resonances.

Table 2 outlines the multimodal resonance frequency of the RPDEA across various prestretch levels, comparing both simulated and experimental results. The deviation between the model and the experiment is small, with a maximum error of 4.5 Hz, ensuring an accuracy rate of over 95%.

In both modal 1 and modal 2, owing to the stiffness effect produced by the Mises stress detailed in Table 3, the resonant frequency rises as the *λ*_2_ increases. Conversely, in modal 3, the resonant frequency does not consistently rise with an increase in *λ*_2_. Simulations indicate that once *λ*_2_ surpasses 1.2, the resonant frequency of modal 3 undergoes marginal change. Moreover, in case *λ*_2_ = 1.3, the resonant frequency of modal 3 drops below that of modal 2, which contrasts with the trend observed under *λ*_2_ equals to 1.1 and 1.2.

Subsequently, we delved into the experimental study on the influences of *λ*_2_ on the multimodal resonance amplitude. According to Figure 4 and Figure 6, the resonance amplitude in backward sweep is close to that in the fixed frequency. Therefore, the comparison among these three RPDEAs is focused on the backward-sweep results. By comparing backward-sweep results shown in Figure 4 and Figure 8, a discernible trend emerges that the amplitude of both superharmonic and harmonic responses of modal 1 resonance decrease as the prestretch increases. Analogous to its suppression on higher-order resonances, this phenomenon also underscores the negative effect of prestretch on the first-order resonance.

Moreover, Figure 4 and Figure 8 demonstrate that the subharmonic resonance yields the largest amplitude. And thus, the subsequent tests only focus on this aspect, as displayed in Figure 10. For modal 1 and modal 2 resonances, the amplitude variations are subtle across different prestretch levels, with *λ*_2_ = 1.2 delivering the optimal performance. However, the amplitude of modal 3 significantly decreases with increasing *λ*_2_. When *λ*_2_ rises to 1.3, the amplitude becomes zero due to the non-excitability.

#### 3.2.2. The Excited Signal

The multimodal resonance of the RPDEA is achieved by the periodically varying Maxwell stress which is controlled by the excited signal shown in Equation (2). The Maxwell stress $\sigma_{M}$ applied on the DE membrane surface can be described as
(3)$$\sigma_{M}=\varepsilon_{r}\varepsilon_{0}{\left(\frac{V_{in}}{T}\right)}^{2}$$
where $\varepsilon_{r}$ and $\varepsilon_{0}$ are the relative permittivity and the absolute permittivity of vacuum of the DE, respectively, and $T=T_{0}/(\lambda_{1}\lambda_{2})$, is the thickness of the DE membrane. By substituting Equation (2) into Equation (3), there is
(4)$$\sigma_{M}=\varepsilon_{r}\varepsilon_{0}E^{2}(\frac{3}{8}+sin2\pi \Omega t-\frac{1}{8}cos4\pi \Omega t){\left(\frac{\lambda_{1}}{\lambda_{1p}}\frac{\lambda_{2}}{\lambda_{2p}}\right)}^{2}$$

If we ignore the variation in *λ*_1_ and *λ*_2_ during the deformation, the Maxwell stress has a partly constant pressure $\sigma_{MC}=3\varepsilon_{r}\varepsilon_{0}E^{2}/8$, applied on the membrane surface, leading to a ‘softness effect’ [20,22]. Thus, increasing *E* will reduce the resonant frequency. Conversely, the increase in *E* can obtain a higher amplitude of Maxwell stress, $\sigma_{M}$, increasing the resonant amplitude. To demonstrate this, we carried out the experiment with varied E. In addition, the simulation about the influence of the constant Maxwell stress $\sigma_{MC}$ on the resonant frequency were carried out, where $\varepsilon_{r}$ is set as 2 [35,36]. These results are shown in Figure 11a–c.

Due to the adverse effect of prestretch on the excitation of higher-order modals, Figure 11a–c exclusively present results under *λ*_2_ = 1.1. The resonant frequency for all experiments was determined by combining both the laser displacement sensor and high-speed camera, with adjustments made manually to the fixed signal frequency. The ‘softness effect’ of $\sigma_{MC}$, as illustrated in Figure 12, shows that the S11 Von Mises stress reduces as *E* increases. The stress can lead to the stiffness of DEA decreases [20,22]. Consequently, as the *E* rises, the resonance frequency of all these three modes exhibits a slight decline.

It is a huge work to develop a specialized FEM model to analysis the multimodal resonance amplitude of RPDEA. To simplify, we use another approach: (a) for modal 1, the sinusoidal maxwell stress equals to $\varepsilon_{r}\varepsilon_{0}E^{2}sin2\pi \Omega t$ is applied on the area shown in Figure 12, while ignoring ${-\varepsilon }_{r}\varepsilon_{0}E^{2}cos4\pi \Omega t/8$; (b) for modal 2 and modal 3, a moment Tsin2πΩt, which is determined by manually changing until it is the same to one of the experimental amplitudes, is applied on the vibrator center, which contacts with the membrane. Ω is same as the corresponding harmonic resonant frequency obtained from FEM. In addition, as the viscoelasticity of the membrane is similar to the damping of fluid [24], the equivalent damping coefficient is proportional to the amplitude. Thus, the damping coefficient in the ‘Modal dynamics’ step increases when E increases. The Maxwell stress, moment, and damping coefficient are listed in Table 4. It should be pointed out that, in the same way as the maxwell stress, the moment amplitude T used in modal 2 and 3 is also proportion to $E^{2}$.

As Figure 11d–f show, both the experiment and the FEM emphasize that a higher *E* results in higher amplitudes for the three modal resonances. And the trending difference between Figure 11e and f indicates that modal 3 possesses greater potential for rotational amplitude than modal 2. This behavior may be attributed to two primary factors: (i) The compliant electrodes are strategically distributed in a lengthwise fashion, leading to a reduced amplitude of periodic stress along the width. (ii) The structural parameter pertaining to width is smaller than that for length.

According to Table 4, although the damping coefficients are close, the applied moment in modal 3 is much higher than that in modal 2, suggesting that the former can output a higher moment. This is an advantage of modal 3. In addition, if we simplify these three resonances to sinusoidal vibrations, their elastic potential energy peaks are estimated to be 1400 × 10^−9^ J, 140,000 × 10^−9^ J, and 1,300,000 × 10^−9^ J under *E =* 50 MV/m. This also indicates that the high-order resonance has more potential in applications.

### 3.3. The Robot Driven by Multimodal Resonance

The above results confirm the multimodal resonance of the RPDEA. The first-order resonance represents vertical vibration, which has been verified to drive a soft-bristled robot [37]. The actuation principle is that the vertical vibration can result in the periodic deflection and relaxation of all the bristles in a phase, leading to the robot sliding or skipping directionally. However, a question remains: can the higher-order resonances also propel robots with soft bristles? To explore this, we assemble a robot with four bristles which are crafted using ELASTOSIL RT 622 silicone (from Wacker Chemie AG). These bristle legs have a length of 5 mm, a thickness of 1 mm, and incline at an angle of 60°.

Inspired by the soft-bristled robot driven by the vertical vibration [38,39], we propose a new mechanism for the rotation vibration of modal 3, as shown in Figure 13a. In the first half cycle, the mass bias swings in a clockwise direction, leading to the extension of the front bristles, but the rear bristles deflect. For the high forward friction of the rear bristles to suppress backward movement, the robot body moves forward for a little displacement. Conversely, in the latter half cycle, owing to the counterclockwise rotation of the mass bias, the former bristles deflect, while the rear bristles extend. The former bristles’ tips become the sticking points for the robot body to incline forward. Finally, after one cycle, the robot moves forward for a net displacement. This actuation principle is verified by the observation obtained from the high-speed camera shown in Figure 13b. Accordingly, the robot is demonstrated to crawl in a hopping mode under modal 3 resonance, where the deflection of the front two bristles is asynchronous to that of the rear two bristles. It should be pointed out that there exists a difference between Figure 13a,b: during the first half, the front bristles can hardly lift off the ground, which may be caused by the lower bending moment (the distance between the mass bias and front bristles is shorter than that between the mass bias and rear bristles). The actuation principle for modal 2’s resonance is similar to that for the third-order resonance, where the periodic deformation happens in the right–left bristles instead of the front–rear bristles.

Figure 14a displays the structure of the robot with four bristles. Figure 14b–d exclusively present the crawling performance driven by the subharmonic resonance, chosen for its superior amplitude. At *E* = 65 MV/m, the maximum velocities achieved by modal 1, modal 2, and modal 3 resonances are 0.94 mm/s, 12.55 mm/s, and 29.3 mm/s, respectively. This indicates that higher-order modals offer superior driving potential for soft-bristled robots in comparison with the first-order modal. An immediate difference, illustrated in Figure 14d, is that higher modals achieve greater positive instantaneous speeds. Note that the speed period is double that of the excitation signals, confirming the subharmonic resonance.

Furthermore, Figure 14c displays robot performance at *E* = 55 MV/m. Note that modal 1 resonance is absent under this case, implying its inability to drive the robot. This may be attributed to the dampening effect of the soft bristles. Additionally, the gradient of the displacement–time curve for modal 2 and 3 at *E* = 55 MV/m is steeper than that for modal 1 at *E* = 65 MV/m, emphasizing the advantage of higher-order resonances.

It should be pointed out that, when the prestretch went up to 1.0 × 1.2 or 1.0 × 1.3, the multimodal resonances cannot drive the robot; that is, we hardly observe the multimodal resonance during the excitation, which needs to be addressed in future work.

## 4. Conclusions

In this study, we introduced an RPDEA which exhibited multimodal resonance when subjected to periodic Maxwell stress. Initially, a simplified FEM simulation was employed to uncover its multimodal resonances: the first-, second-, and third-order resonances corresponded to vertical vibrations, rotations along the length, and rotations along the width, respectively. These findings were subsequently validated through dynamic response tests conducted on an RPDEA with a prestretch of 1.0 × 1.1. The experiments illuminated that, by modulating the excited frequency, it was possible to elicit superharmonic, harmonic, and subharmonic resonances for the first order. However, for the second and third orders, only harmonic and subharmonic responses could be activated.

By comparing the dynamic responses, it was found that the resonant frequencies of both the first order and the second order modals ascended with increasing prestretch. Conversely, the experiments indicated that the influence of prestretch on the amplitudes of these two resonances were small. In addition, the heightening prestretch appeared to inhibit the third-order resonance, such as the absence of the third-order resonance once the prestretch went up to 1.0 × 1.3. 

In the combination of the experiment and the FEM, when elevating the nominal electric field intensity under the prestretch of 1.0 × 1.1, the resonant frequency saw a slight decline, while the amplitude occurred a noticeable surge in these three different order resonances. And the elastic potential energy in their stable state increased with the order increased.

Lastly, by merging RPDEA with soft bristles, we confirmed a new actuation mode for driving the soft-bristled robot. This suggested that the rotation vibration could lead to a directionally hopping motion, as demonstrated by a high-speed camera, where the front two bristles and the rear two bristles deflected in different phases. And the crawling performance test showcased the efficacy of the rotation vibration (second- and third-order resonances) in propelling the robot directionally. Remarkably, the third-order resonance exhibited the greatest potential among these three vibrations, serving as a practical example for the advantages of higher-order resonance. And this higher-order resonance with rotation may hold promise for the innovation of swimming or flying robots. In addition, a limitation of this work is that we could not achieve a comprehensive simulation of the multimodal resonance; this will be addressed in future work.

## Figures and Tables

**Figure 1 biomimetics-09-00488-f001:**
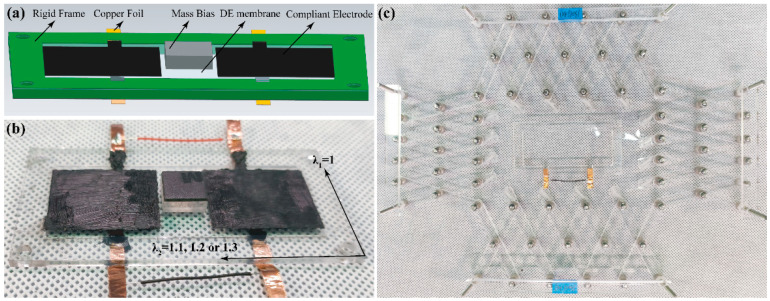
(**a**) The schematic diagram of RPDEA; (**b**) the physical body; (**c**) the setup for the prestretching DE membrane.

**Figure 2 biomimetics-09-00488-f002:**
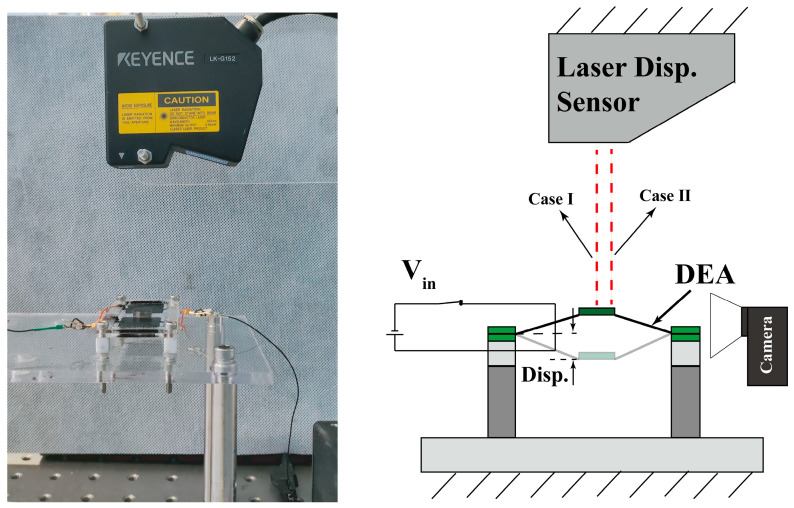
The experimental setup for dynamic response test of the RPDEA.

**Figure 3 biomimetics-09-00488-f003:**
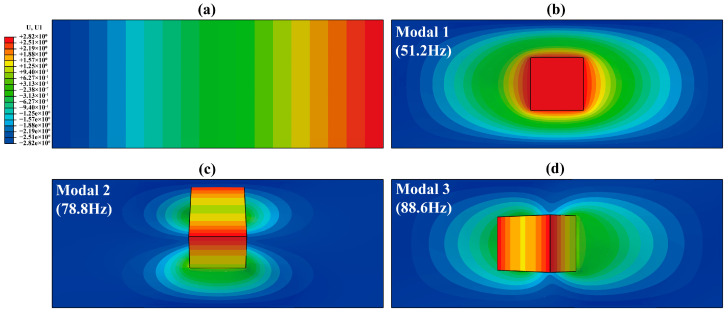
(**a**) The deformation in length of the silicone membrane without Maxwell stress under prestretch of 1.0 × 1.1. (**b**) The first-order resonance (modal 1). (**c**) The second-order resonance (modal 2). (**d**) The third-order resonance (modal 3).

**Figure 4 biomimetics-09-00488-f004:**
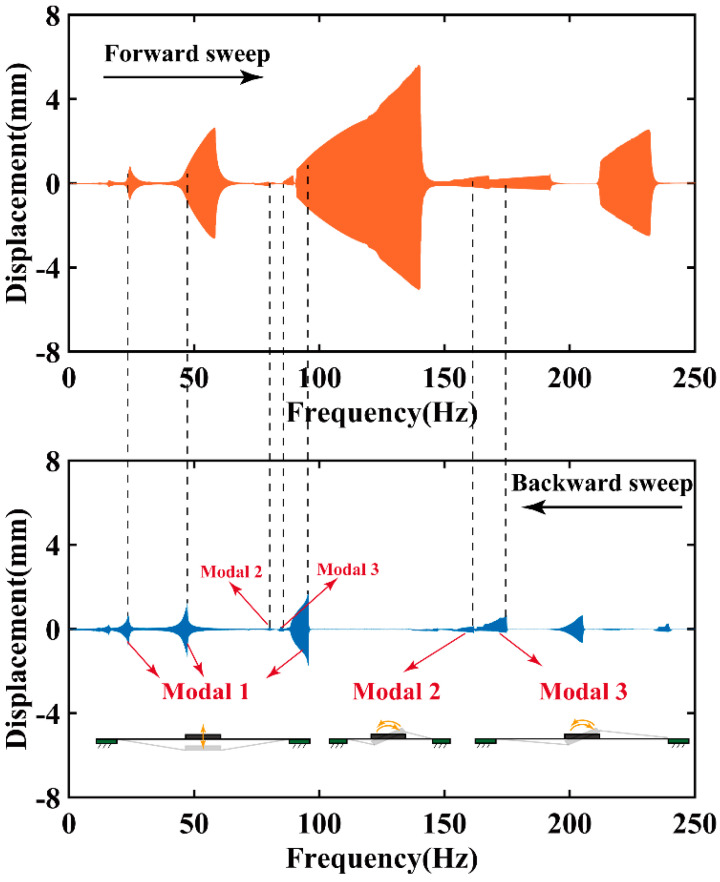
The forward and backward sweep test under both Part A and Part B conducted synchronously, prestretch *λ*_1_
*× λ*_2_ = 1.0 × 1.1, and *E* = 45 MV/m.

**Figure 5 biomimetics-09-00488-f005:**
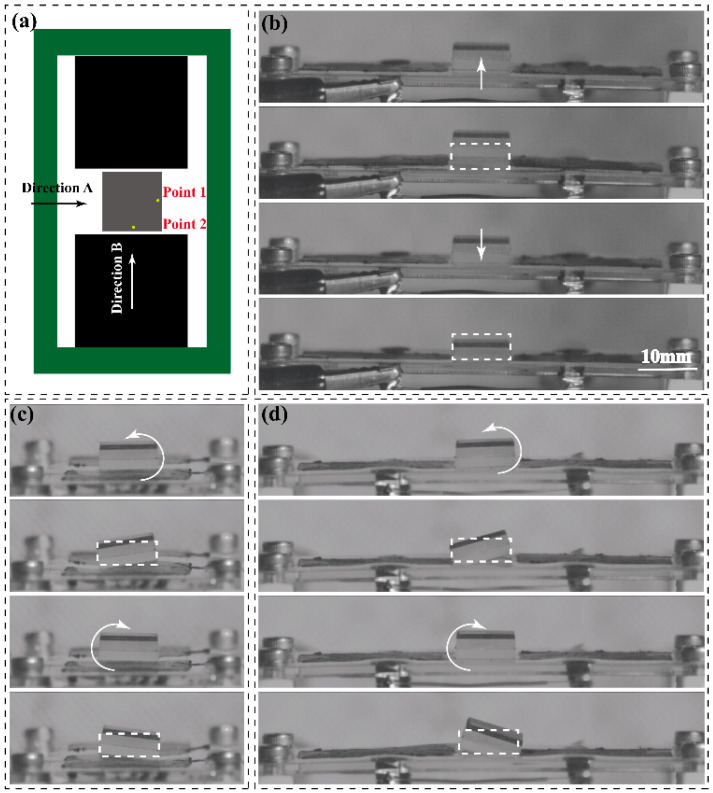
The snapshots of the RPDEA with prestretch of 1.0 × 1.1. (**a**) The view direction of high-speed camera (Revealer 2F04C, from AgileDevice, Hefei, China, 1250 FPS): (**b**,**d**) with direction A, while (**c**) with direction B. (**b**) *E* = 50 MV/m, *Ω* = 96 Hz (Appendix A); (**c**) *E* = 45 MV/m, *Ω* = 156 Hz (Appendix A); (**d**) *E* = 40 MV/m, *Ω* = 175 Hz (Appendix A). The dotted boxes refer to the original position.

**Figure 6 biomimetics-09-00488-f006:**
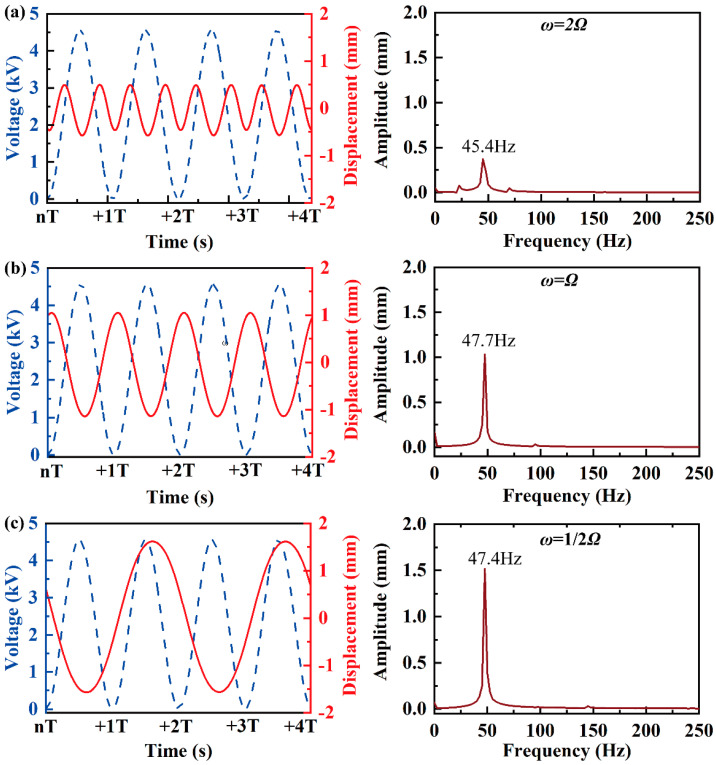
The fixed-frequency dynamic response of three peak in modal 1 resonance shown in Figure 4 under *E*= 50 MV/m (the left shows the response vs. time, and the right shows the DFT results). (**a**) *ω* = 23 Hz (superharmonic response); (**b**) *ω* = 47 Hz (harmonic response); (**c**) *ω* = 96 Hz (subharmonic response).

**Figure 7 biomimetics-09-00488-f007:**
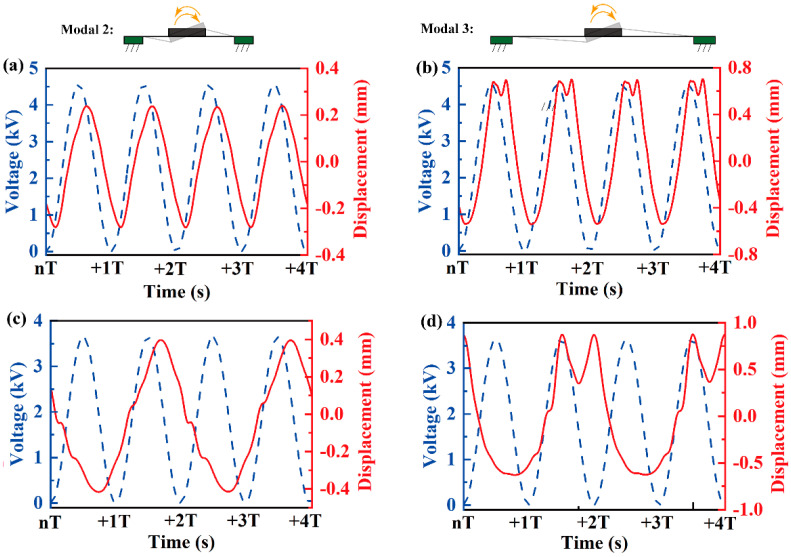
The fixed-frequency dynamic response of modal 2 and modal 3 resonance shown in Figure 4; (**a**) modal 2, *Ω* = 78 Hz, under *E* = 50 MV/m (harmonic response); (**b**) modal 3, *Ω* = 87 Hz, under *E* = 50 MV/m (harmonic response); (**c**) modal 2, *Ω* = 158 Hz, under *E* = 40 MV/m (subharmonic response); (**d**) modal 3, *Ω* = 175 Hz, under *E* = 40 MV/m (subharmonic response). For modal 2, the light-spot of the laser displacement sensor is located at point 1 in Figure 5a; while for modal 3, it is located at point 2. The yellow arrows represent the rotation vibration.

**Figure 8 biomimetics-09-00488-f008:**
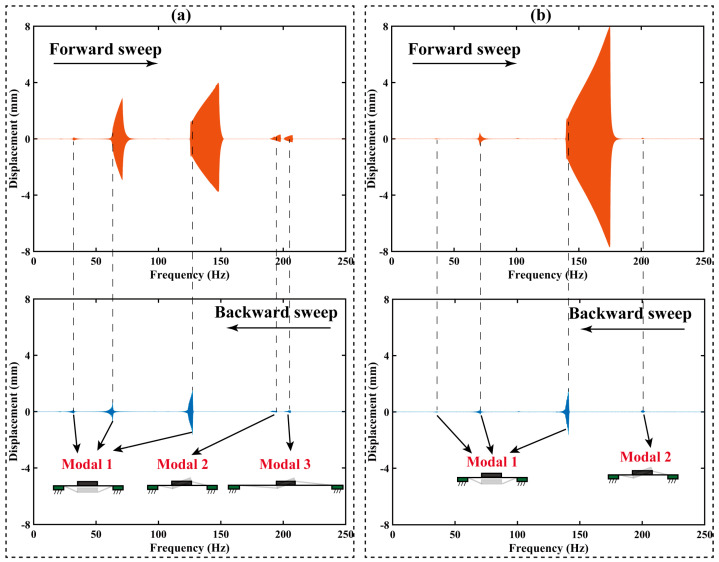
The forward and backward sweep test under *E* = 45 MV/m, (**a**) *λ*_1_ × *λ*_2_ = 1.0 × 1.2, (**b**) *λ*_1_ × *λ*_2_ = 1.0 × 1.3.

**Figure 9 biomimetics-09-00488-f009:**
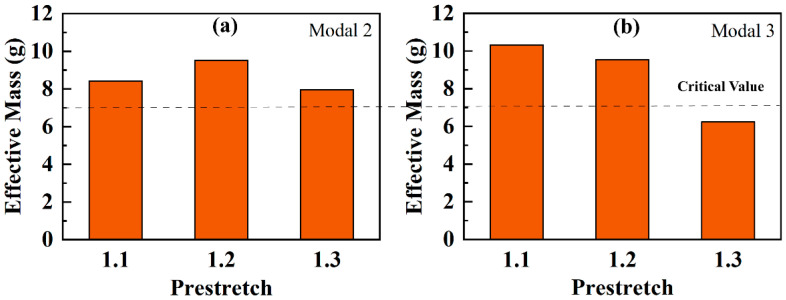
The effective mass in modal 2 (**a**) and modal 3 (**b**) of prestretch *λ*_2_ = 1.1, 1.2, and 1.3, obtained from ‘Job monitor’ in FEM software.

**Figure 10 biomimetics-09-00488-f010:**
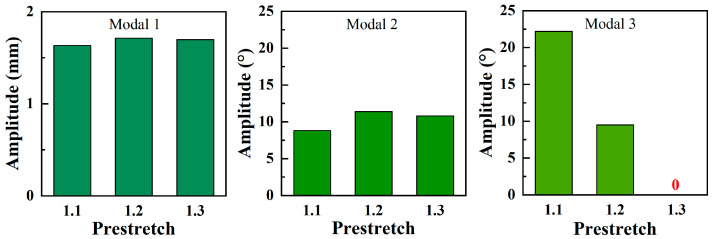
The subharmonic resonant amplitude of the prestretch *λ*_2_ = 1.1, 1.2, and 1.3 (*E* = 50 MV/m). The red ‘0’ represents ‘non-excitability’.

**Figure 11 biomimetics-09-00488-f011:**
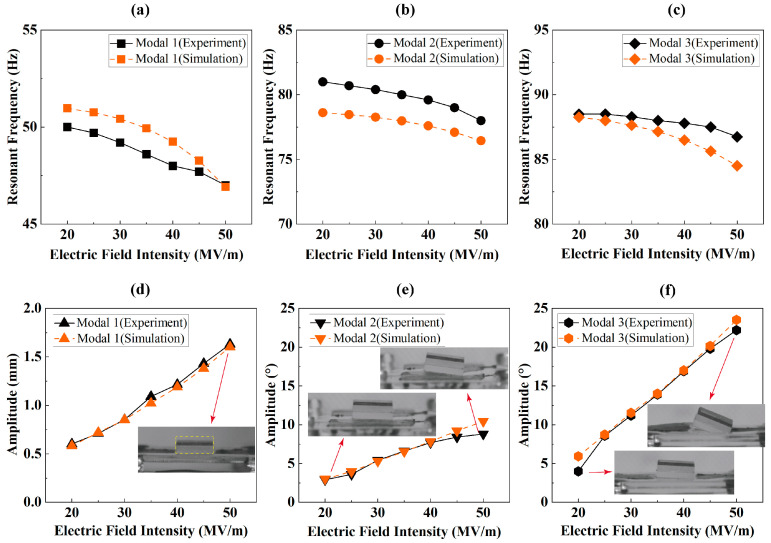
The resonant frequency of the response and the superharmonic resonance amplitude of the prestretch *λ*_2_ = 1.1 under different *E*. The yellow dotted box represents the original position of the vibrator. (**a**,**d**) modal 1; (**b**,**e**) modal 2; (**c**,**f**) modal 3.

**Figure 12 biomimetics-09-00488-f012:**
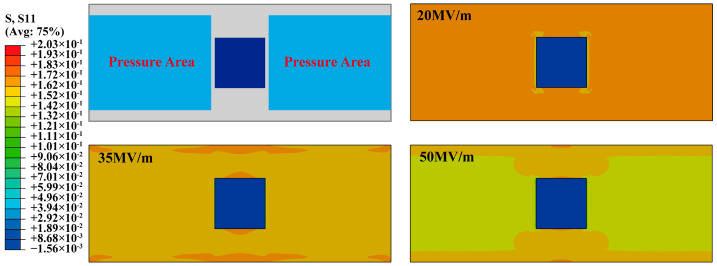
The Mises stress in S11 of the RPDEA under different $\sigma_{MC}$ (Equation (4)).

**Figure 13 biomimetics-09-00488-f013:**
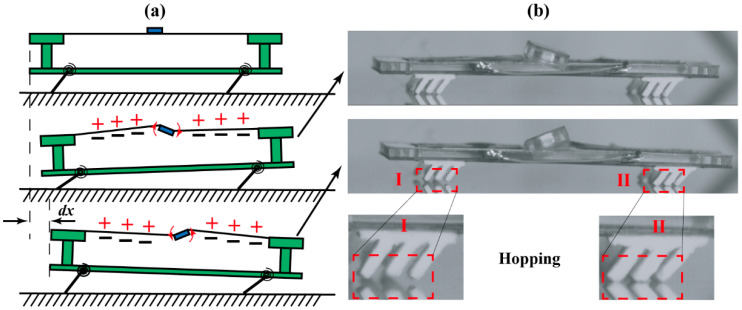
The actuation principle for soft-bristled robot under the third-order resonance. (**a**) Schematic diagram. (**b**) Image from the high-speed camera (Appendix A).

**Figure 14 biomimetics-09-00488-f014:**
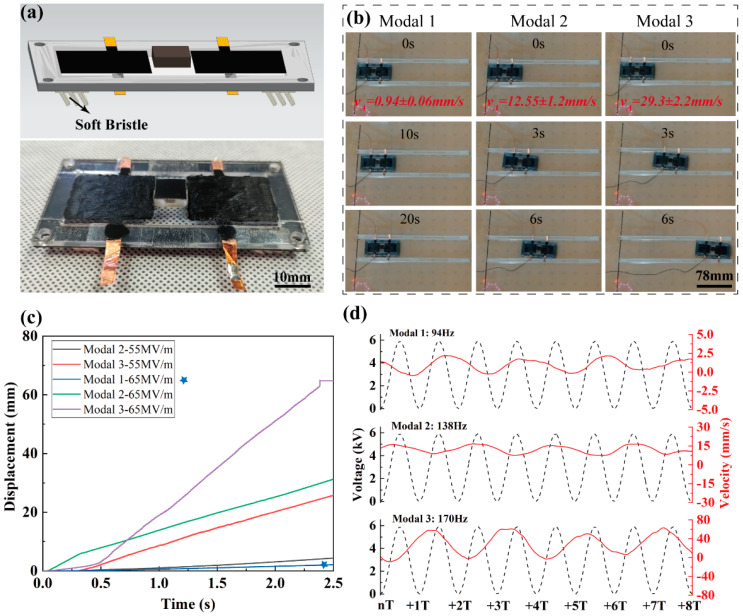
(**a**) The structure of the soft-bristled robot; (**b**) performance images of modal 1 (Appendix A), modal 2 (Appendix A), and modal 3 (Appendix A) under *E* = 65 MV/m; (**c**) the result tested by displacement sensor; (**d**) the instantaneous velocities of modal 1, modal 2, and modal 3 under *E* = 65 MV/m. The blue star represents ‘Modal 1-65MV/m’.

**Table 1 biomimetics-09-00488-t001:** The parameters of Yeoh model used for the FEM.

Parameter	*C* _1_	*C* _2_	*C* _3_	*D* _1_	*D* _2_	*D* _3_
Value	0.22	−0.08/−0.09	0.05	0.2	0	0

**Table 2 biomimetics-09-00488-t002:** The multimodal resonant frequency of different prestretch under *E* = 45 MV/m.

Frequency (Hz)	1.0 × 1.1		1.0 × 1.2		1.0 × 1.3	
	Simulation	Experiment	Simulation	Experiment	Simulation	Experiment
Modal 1	48.3	47.7	63.1	63.0	70.1	73.2
Modal 2	77.1	79.0	95.0	96.3	101.6	100.6
Modal 3	85.6	87.5	98.0	102.5	98.3	/

**Table 3 biomimetics-09-00488-t003:** The Mises stress (FEM simulation) of the silicone membrane.

Prestretch	1.0 × 1.1	1.0 × 1.2	1.0 × 1.3
Mises Stress (MPa)	0.14	0.26	0.36

**Table 4 biomimetics-09-00488-t004:** The Maxwell stress, moment, and damping coefficient used in Figure 11d–f.

*E* (MV/m)		20	25	30	35	40	45	50
Modal 1	Maxwell stress (10^−2^ Mpa)	0.71	1.11	1.59	2.17	2.83	3.58	4.43
	Damping (10^−2^ N·s/mm)	0.24	0.30	0.36	0.40	0.44	0.47	0.49
Modal 2	Moment (10^−3^ N·mm)	0.17	0.27	0.38	0.52	0.68	0.86	1.04
	Damping (10^−2^ N·s/mm)	0.24	0.3	0.33	0.37	0.41	0.45	0.49
Modal 3	Moment (10^−3^ N·mm)	0.58	0.90	1.30	1.76	2.30	2.92	3.6
	Damping (N·s/mm)	0.27	0.3	0.33	0.38	0.42	0.47	0.52

## Data Availability

The data supporting reported results can be made available via requesting with the corresponding author.

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
