# Peer review of "Multimodal Resonances of a Rectangular Planar Dielectric Elastomer Actuator and Its Application in a Robot with Soft Bristles"

_biomimetics, 2024, doi:10.3390/biomimetics9080488_

Round 1
Reviewer 1 Report
Comments and Suggestions for Authors
The paper is devoted to the description of three resonance modes of vibrations of rectangular silicone dielectric structures arising under the application of electric voltages. The methods of experimentation and numerical modelling are described in detail, the reliability of the results obtained is not in doubt. The topic is actual, the authors considered a specific structure based on polymer membranes of a certain shape and showed the possibility of its motion.
Avantage:
A detailed description of the structure motion process in the third resonant mode.
Questions:
1) Maxwell stress is stated as the root cause of oscillations. Namely, an alternating voltage is applied to a sample coated with a conductive layer. The sample is a dielectric. What do cause the deformation of the sample?
2) In Figure 6, the resonance frequencies at the main resonance frequency, subharmonics and superharmonics are indicated on the right-hand side. However, the amplitude-frequency dependences show resonances at approximately the same frequency, although multipled frequencies are indicated in the captions. Could the authors please explain in more detail what kind of dependencies and values are presented here?
3) Note: in the diagram of Fig. 2b it is signed that the circuit is AC voltage, with the designation corresponding to DC voltage.
Author Response
Comment 1. Maxwell stress is stated as the root cause of oscillations. Namely, an alternating voltage is applied to a sample coated with a conductive layer. The sample is a dielectric. What do cause the deformation of the sample?
Answer: Thanks for the comment. The Maxwell stress, which caused by attraction force of positive and negative charges located at the two sides of dielectric elastomer respectively, is periodic under the sinusoidal voltage. The periodic Maxwell stress results in the thickness of the elastomer shrinks in cycle, and corresponding out-of-plane deformation of RPDEA. If the shrinkage frequency closes to the inherent resonant frequency of the elastomer, a significant deformation (resonance) will occur. The revision can be found in Page 3, Paragraph 1, lines 98-102.
Comment 2. In Figure 6, the resonance frequencies at the main resonance frequency, subharmonics and superharmonics are indicated on the right-hand side. However, the amplitude-frequency dependences show resonances at approximately the same frequency, although multipled frequencies are indicated in the captions. Could the authors please explain in more detail what kind of dependencies and values are presented here?
Answer: Thanks for the comment. With the excited frequency goes up twofold (from superharmonic to subharmonic resonance), the input power goes up, resulting in their resonant amplitude ascend. However, because of the nonlinear damping of dielectric elastomer and structural nonlinearity of PRDEA, the relationship between the input frequency and resonant amplitude is nonlinear. The revision can be found in Page 7, Paragraph 2, the last sentence.
Comment 3. Note: in the diagram of Fig. 2b it is signed that the circuit is AC voltage, with the designation corresponding to DC voltage.
Answer: Thanks for the comment. In Figure 2b, the ‘AC voltage’ is modified to ‘Vin’ which refers to Equation (2), Page 4.
Reviewer 2 Report
Comments and Suggestions for Authors
1. The vibration mode should be a range of frequency, instead of a single frequency, as shown in the sweep test.
2. The reasons of different vibration mode frequencies and amplitude need to to be further articulated.
Author Response
Comment 1: The vibration mode should be a range of frequency, instead of a single frequency, as shown in the sweep test.
Answer: Thanks for the comment. Due to the viscoelasticity, each order modal of RPDEA is in a certain frequency range, where their resonant behavior is consistent. In practical applications, such as soft robots, we prefer to use the maximum resonance to maximize robot performance. Therefore, in this work we focused on the maximum amplitude of each modal resonance. The corresponding addition can be found in Page 4, lines 152-155.
Comment 2: The reasons of different vibration mode frequencies and amplitude need to be further articulated.
Answer: Thanks for this comment. For modal 1, we further analyzed the reason for the difference among the superharmonic, harmonic and subharmonic resonance. This revision can be found in Page 7, lines 223-225. To more clearly describe the influence mechanism of the field intensity on the resonant frequency, we added a sentence ‘The stress can lead to the stiffness of DEA decreases [20,22]’ in Page 11, line 345. And we used ‘the moment amplitude T’ instead of ‘the moment’ to analyzed the effect of field intensity on the amplitude of Modal 2 and Modal 3 resonance, in Page 11, line 359. There is a regret that, at current we cannot overall conduct the multimodal resonance amplitude simulation due to the complexity of RPDEA deformation perfectly, which will be addressed in future work.
Reviewer 3 Report
Comments and Suggestions for Authors
In this manuscript, the authors combine experimental and finite element modeling to scrutinize the multimodal resonance of a rectangular planar DEA (RPDEA) with a central mass bias. Both experimental and finite element results provide good evidence that the frequency and amplitude of the multimodal resonance can be tuned by adjusting the amplitude of the excitation signal. The authors also found that rotational vibrations can be used instead of linear vibrations to drive a soft-bristle brush robot in a jumping motion, which results in the robot moving faster than the first-order resonance. This paper is well researched, and the experiments and finite element simulations adequately illustrate the paper's conclusions, and it is recommended for publication.
Some minor formatting issues need to be corrected. For example, the abbreviations DC and AC appear for the first time in the abstract and it is recommended that they be corrected to their full names.
Author Response
Comment 1: Some minor formatting issues need to be corrected. For example, the abbreviations DC and AC appear for the first time in the abstract and it is recommended that they be corrected to their full names.
Answer: Thanks for the comment. In abstract, the full names of ‘DC’ and ‘AC’ are ‘direct current’ and ‘alternating current’, respectively. This has been revised in Page 1, ‘Abstract’ part, lines 25-26. We also examined the full text in detail, correcting ‘CAD model’ into ‘computer aided design’ and ‘FPS’ into ‘frames per second’, in Page 3, Lines 120 and in Page 5, Line 174 respectively. Besides, in line 212, ‘these resonance’ is corrected to ‘these resonances’.